# Effectiveness, Tolerability, and Safety of Tofacitinib in Rheumatoid Arthritis: A Retrospective Analysis of Real-World Data from the St. Gallen and Aarau Cohorts

**DOI:** 10.3390/jcm8101548

**Published:** 2019-09-26

**Authors:** Ruediger B. Mueller, Caroline Hasler, Florian Popp, Frederik Mattow, Mirsada Durmisi, Alexander Souza, Paul Hasler, Andrea Rubbert-Roth, Hendrik Schulze-Koops, Johannes von Kempis

**Affiliations:** 1Division of Rheumatology and Immunology, Department of Internal Medicine, Kantonsspital St. Gallen, 9007 St. Gallen, Switzerland; fmpopp@gmail.com (F.P.); fmattow@googlemail.com (F.M.); andrea.rubbert-roth@uk-koeln.de (A.R.-R.); johannes.vonkempis@kssg.ch (J.v.K.); 2Division of Rheumatology, Medical University Department, Kantonsspital Aarau, 5001 Aarau, Switzerland; carolinehasler4@gmail.com (C.H.); m.durmisi@stud.unibas.ch (M.D.); paul.hasler@ksa.ch (P.H.); 3Division of Rheumatology and Clinical Immunology, Department of Internal Medicine IV, Ludwig-Maximilians-University Munich, 80336 Munich, Germany; Hendrik.Schulze-Koops@med.uni-muenchen.de; 4Iterata AG, 5722 Gränichen, Switzerland; souza@iterata.ch

**Keywords:** tofacitinib, rheumatoid arthritis, oral

## Abstract

**Introduction:** Tofacitinib is an oral JAK inhibitor indicated for the treatment of rheumatoid arthritis (RA). The efficacy and safety of tofacitinib have been shown in several randomized clinical trials. The study presented here aimed to assess the clinical tolerability and effectiveness of tofacitinib among RA patients in real life. **Methods:** Consecutive patients between January 2015 and April 2017 with RA who fulfilled the American College of Rheumatology (ACR)/European League Against Rheumatism (EULAR) 2010 criteria were included in a prospectively designed analysis of retrospective data. Patients were initiated on tofacitinib 5 mg bid. The primary objective was to analyze the safety of tofacitinib in a real-life cohort. Safety was assessed by the reasons to stop tofacitinib during follow up and changes of liver enzymes, hemoglobin, and creatinine. The secondary outcome was to analyze the frequency of and time to achieve low disease activity (LDA) and remission as defined by 28 joint count disease activity score (DAS28). **Results:** A total of 144 patients were treated with tofacitinib. A total of 84.9% of patients were pre-exposed to at least one biological agent. The average DAS28 at the initiation of tofacitinib was 4.43. A total of 50.0% of patients were positive for rheumatoid factor and 49.0% for ACPA. The mean follow up was 1.22 years (range 10d–3.7a) after initiation of tofacitinib treatment. A total of 94 (64.4%) patients remained on tofacitinib during follow-up. The average time to stop tofacitinib was 190.0 days. Reasons to stop tofacitinib were: insufficient response (*n* = 23), gastrointestinal symptoms (*n* = 18), infection (*n* = 5), myalgia (*n* = 2), remission (*n* = 2), headache (*n* = 2), cough, blue finger syndrome, intolerance, heartburn, psoriasis, and increased liver enzymes (all *n* = 1). Increased alanine amino transferase (ALAT) or aspartate amino transferase (ASAT) > 2× upper limit of normal (ULN) were detected in 3.3% and 4.4% of patients, respectively. Hemoglobin decrease of >10% was detected in 15.1% of the patients and decreased lymphocytes <500/μL in 3.4%. An increase of creatinine >20% was detected in 9.4% of patients. A total of 62.9% and 50.0% of the patients achieved low disease activity (LDA) or remission after a median of 319 and 645 days, respectively. These rates were significantly higher in patients naïve to biologic agents as compared to patients pre-exposed to biologics (LDA: naïve 100% 92 d, pre-exposed 57.0% 434 d, *p* ≤ 0.001; remission: naïve 86.7% 132 d, pre-exposed 44.1%, 692 d, *p* = 0.001). **Conclusions:** Tofacitinib is a safe and effective treatment option for patients with RA. Tofacitinib may induce high rates of LDA and remission in patients with active disease, even after the use of one or more biologics, though the rate appeared higher in patients naïve to biologics. Tofacitinib may be a valuable option in a treat-to-target approach. Our data demonstrate that Janus kinase (JAK) inhibitors are safe and efficacious in real life patients.

## 1. Introduction

Rheumatoid arthritis (RA) is a chronic autoimmune disease characterized by the inflammation and destruction of joints. It may result in functional impairment, declining health status and reduced quality of life for affected patients [1,2,3]. The principal goal in the treatment of RA is to achieve and maintain remission, or, if that is not attainable, low disease activity (LDA) [4,5].

Conventional synthetic (cs) disease-modifying anti-rheumatic drugs (DMARDs), especially methotrexate (MTX), have long been the cornerstones of RA treatment. In the last 20 years, biologic agents have broadened the clinical armamentarium [6]. Though biologics have revolutionized the managing of RA [7,8,9,10,11,12,13,14,15,16,17,18,19,20,21,22,23,24], their effects are limited. Approximately 50% of RA patients treated with biologics meet the criteria for low disease activity (28 joint count disease activity score (DAS28) ≤ 3.2) or remission (DAS28 < 2.6), while a significant proportion of patients do not achieve an ACR 20 (American College of Rheumatology) response [14,15]. Furthermore, patients on biologics may experience adverse events (AEs) or loss of effectiveness over time [25], e.g., by developing anti-drug antibodies. To quantify the unmet need for additional therapies, Drosos et al. performed a long-term, real-world observational study of their cases with RA treated according to the European League Against Rheumatism (EULAR) and American College of Rheumatology (ACR) recommendations. Approximately one-fifth of their patients did not respond sufficiently to csDMARDs or bDMARDs (biological disease-modifying anti-rheumatic drugs), substantiating the need for alternative treatments [26].

Tofacitinib is a novel, oral Janus kinase (JAK) inhibitor indicated for the treatment of RA. JAK inhibitors are small-molecule drugs that interfere with the activation of JAKs, a family of enzymes implicated in the signaling of leukocytes. JAK signaling has been shown to play an essential role in immune cell generation, differentiation and responses [27,28,29]. By inhibiting these signaling mechanisms, JAK inhibitors such as tofacitinib have the potential to successfully interfere with immune activation that is critical for RA [30,31,32] (Koehler, J. Clin. Med. 2019, 8, 938).

Phase II and III clinical trials have shown that the treatment of RA patients with tofacitinib, either as a monotherapy or in combination with csDMARDs, is capable of significantly reducing disease activity, as measured by ACR response rates, EULAR responses and HAQ-DI scores [33,34,35,36,37,38,39]. Studies comparing tofacitinib to other therapeutic strategies in the treatment of RA suggest that the effectiveness of tofacitinib is similar to that of biologic agents [40,41,42,43]. The safety profile of tofacitinib does not appear to differ significantly from biologics [34,39,40,41,42,43,44].

In 2012 and 2014, the FDA and the Swissmedic approved tofacitinib for adult patients with moderate to severe RA who had a prior inadequate response to MTX. Approval from the European Medicines Agency (EMA) was granted in 2017. With JAK inhibitors still representing a relatively novel treatment option in the management of RA, there is a demand to use the experience gained through using tofacitinib in a real-life, clinical setting, to further evaluate its safety and utility. In this study, we aimed to analyze real-life data from routine clinical practice to compare our experience with the results of controlled studies.

## 2. Methods

### 2.1. Patient Recruitment

For this retrospective analysis of data, patients were recruited through a chart review of all RA patients at the hospitals of St. Gallen and Aarau, Switzerland. Patients with a clinical diagnosis of RA consistent with the current definition in the 2010 ACR/EULAR criteria were required [45] and initiation of oral tofacitinib 5 mg bid followed. Exclusion criteria were ages younger than 18 years or older than 80 years at disease onset. All patient charts of the cohort from Aarau and St. Gallen were screened sequentially for eligibility. Thus, selected patients were followed until tofacitinib administration was terminated or until the last visit entered in the database. The decision to stop tofacitinib and all other decisions concerning treatment were at the discretion of the treating clinician. Ethical approval for the collection of patient data was given by the regional review board.

### 2.2. Study Design

This was a longitudinal, retrospective chart review conducted between April 2013 and September 2017 within the St. Gallen and Aarau RA cohorts. The pre-defined primary endpoints were the incidence of adverse events, changes in laboratory values (increase in alanine amino transferase (ALAT) or aspartate amino transferase (ASAT) > 1.2 or 2.0 above the upper limits of normal), decrease in hemoglobin of >10%, lymphocytes <500 or <1000/μL, increase in creatinine >20%, and adverse events leading to the termination of tofacitinib treatment. The pre-defined secondary clinical endpoint was longitudinal disease activity as measured by DAS28 and the achievement of LDA (DAS28 ≤ 3.2) and remission (DAS28 < 2.6). Data were analyzed for the entire cohort of 144 patients, and, as a secondary analysis, separately for patients who had prior exposure to biologic agents and patients who were naïve to biologic agents.

### 2.3. Statistical Methods

Summary statistics are reported as median (range) or *n* (%). Kaplan–Meier curves were plotted, and Kaplan–Meier estimates with 95% confidence intervals based on a log–log transformation were computed for the endpoints. Time to LDA and remission was compared between patients with and without prior exposure to biologics with a log-rank test. All analyses were performed in the R programming language (R Foundation, Vienna, Austria, version 3.3.3, R Core Team 2013).

## 3. Results

### 3.1. Baseline Demographics

A total of 144 patients from the rheumatology units of the St. Gallen and Aarau rheumatology divisions fulfilled the inclusion criteria and were included in the cohort. The mean age at initiation of tofacitinib was 59.7 years and mean disease duration was 9.1 years. The majority of patients were female (69.4%). A total of 50% were positive for rheumatoid factor (RF), and 48.6% were positive for anti-citrullinated protein antibodies (ACPAs), as described in the records. No additional testing for RF and/or ACPA prior or under tofacitinib treatment was conducted. A total of 56% of the patients were either RF and/or ACPA positive.

Disease activity among the patient cohort was moderate, with a mean DAS28 of 4.43 at the initiation of tofacitinib. A total of 63.3% had a disease classified as erosive. All patients were initiated on a baseline dose of tofacitinib 5 mg bid. Regarding other medications, the mean number of previous csDMARDS was 1.9. A total of 84.7% of patients had been previously exposed to at least one biologic agent; the mean number of previous biologics was 2.2. Mean follow-up was 1.22 years (range 10 days–3.7 years) after initiation of tofacitinib (Table 1).

### 3.2. Disease Activity

For all patients, the mean DAS28 decreased significantly from 4.4 at baseline to 3.59, 3.22, 3.18, and 3.13 at 90, 180, 270, and 360 days (Figure 1). In total, 53% of patients achieved LDA and 48% DAS28 defined remission. The median time to LDA and remission was 319 days and 645 days, respectively.

The rates of LDA and remission under tofacitinib were higher in patients naïve to biologics compared to patients who had been previously exposed: 100% of naïve patients achieved LDA, and 83.3% achieved remission, as compared to 53.3% and 44.9% of pre-exposed to biologics patients. Also, the duration of tofacitinib treatment until LDA or remission was shorter in patients naïve to biologics. Patients in this cohort achieved LDA after a median 92 days and remission after a median 132 days, while medians for achieving LDA and remission among patients pre-exposed to biologic agents amounted to 434 days and 692 days, respectively. In both cases, the difference between naïve and pre-exposed patients was statistically significant (Figure 2, *p* < 0.001).

### 3.3. Discontinuation

A total of 89 (61.8%) patients remained on tofacitinib during follow-up. The median time to stop tofacitinib was 95 days (range: 4–1106). A total of 21 patients (14.6%) stopped tofacitinib due to insufficient responses and 35 patients (23.6%) stopped due to adverse events (AEs, Table 2). Of these, the most frequent reasons for discontinuing tofacitinib were gastrointestinal symptoms (*n* = 18), followed by infection (*n* = 5), myalgia (*n* = 2), remission (*n* = 2), headache, cough, blue finger syndrome, intolerance, heartburn, psoriasis, and increased liver enzymes (all *n* = 1). The median time to stop tofacitinib treatment due to ineffectiveness was 204 days (Figure 3). The median time to stop treatment due to AEs ranged from 10 to 290 days (Figure 3). None of the demographic parameters at baseline was a significant predictor for stopping tofacitinib.

### 3.4. Laboratory Values

Laboratory values including liver enzymes, creatinine, lymphocyte count, and hemoglobin were followed during tofacitinib treatment. Increased ALAT or ASAT > 2× ULN were detected in 3.3% and 4.4% of patients, respectively. These changes were transient in 50% and 60% of cases, respectively. Hemoglobin decrease of >10% was detected in 15.1% of patients and decreased lymphocytes <500/μL in 3.4%. An increase in creatinine >20% was detected in 9.4% (Figure 4).

## 4. Discussion

This study retrospectively analyzed real-life data from a cohort of 144 RA patients treated with tofacitinib 5 mg bid, with the aim of assessing the effectiveness and tolerability of tofacitinib in a clinical setting.

### 4.1. Effectiveness

Among the patient cohort, tofacitinib significantly reduced disease activity, with 58.2% of patients achieving LDA and 49.5% achieving remission at follow-up. This is a little higher than in published phase I–III clinical trials. In these clinical trials, the overall proportion of RA patients achieving DAS28 defined as was LDA 5.7%–47.5% [34,37,43,46,47] and remission 7.2%–23.1% [34,36,37,43,46,47,48,49], depending on the exposure to and efficacy of previous treatments. Essentially, our findings corroborate those of previous studies that have shown tofacitinib to be effective in the management of RA [33,34,35,36,37,48,50].

A total of 15.9% of our patients stopped tofacitinib due to ineffectiveness. Percentages of inefficacy were not published in the pivotal clinical trials, especially as this is not a defined outcome. Therefore, the best approximation may be missing an ACR 20 response. The ACR 20 response was not reached in 33.9% of the MTX-IR (methotrexate incomplete responders) patients and 48.2% of the TNF-IR (tumor necrosis factor incomplete responders) patients [51] in the phase II and III program for tofacitinib and 28.7% of naïve patients [47]. In a long-term extension study, 20.4% of patients did not achieve ACR 20 after 24 months and 21.5 after 96 months [52]. Importantly, not achieving ACR 20 does not necessarily mean that a patient or a treating physician considers the therapeutic response to, e.g., tofacitinib, ineffective in a clinical setting. Thus, the rate of 15.9% of patients stopping tofacitinib for ineffectiveness appears to be somewhat lower than observed in the clinical studies and long-term extension studies. However, because, as outlined above, missing an ACR 20 response does not necessarily reflect inefficacy, we think that these rates are comparable.

Although tofacitinib demonstrated effectiveness across all patient demographics, a significant difference was observed between patients naïve to biologic agents and patients who had previously been exposed to biologics: naïve patients had a trend of a higher rate of achieving LDA and remission compared to pre-exposed patients. Also, the duration of tofacitinib treatment until LDA and remission was significantly shorter in patients naïve to biologics. However, the small number of patients naïve to biologics have to be taken into account. These patients naïve to biologics had a shorter mean duration of disease at initiation of tofacitinib and lower mean baseline DAS28, which may have influenced the results. Shorter disease duration [53,54,55] and lower disease activity at initiation of treatment [56,57,58,59] have both been shown to correlate with higher rates of LDA and remission in RA patients. However, the indication that previous biologic therapies are associated with a reduced clinical response to tofacitinib is consistent with recent studies: a meta-analysis of phase II and III clinical trials of tofacitinib in RA patients published in 2016 showed that patients who were naïve to biologics had a numerically better clinical response compared to patients with a prior inadequate response to biologics [60]. This finding was confirmed by a direct comparison study [61] In the 2015, ACR guidelines for the treatment of RA tofacitinib were still recommended as a second-line drug after treatment with biologic agents resulted in an inadequate response or intolerance [62]. In the EULAR guidelines, tofacitinib is already recommended for RA that has inadequately responded to one or more csDMARDs [63].

This is reflected in the patient cohort of the present analysis, in which only 15.2% of patients were biologic-naïve. However, this and other studies suggest that there is a benefit to be gained from using tofacitinib early in the treatment of RA, before the initiation of biologics, which may call into question the position of tofacitinib as a second-line drug.

### 4.2. Adverse Events

Few AEs leading to the discontinuation of tofacitinib treatment were observed in this study. Among the AEs that lead to stopping tofacitinib, the most frequent were gastrointestinal AEs, followed by infections. Patients experienced no severe or life-threatening AEs under tofacitinib. The safety profile was comparable to published data, except for a single case of blue finger syndrome [64]. Following initiation, patients developed an increase in LDL cholesterol, which has been established as a side effect of tofacitinib treatment in previous studies [65,66]; however, a tofacitinib-induced increase in cholesterol does not appear to be associated with a higher incidence of cardiovascular AEs in patients [60,65,67].

It is interesting that AEs are the main reason for stopping tofacitinib in the period directly following initiation of treatment, later superseded by insufficient therapeutic responses. Treatment had to be stopped for AEs, if necessary, usually rather early in the course of treatment, considering that the average time of follow up was 1.22 years and the mean time to stop tofacitinib was 183 days. Most patients (23%) discontinued treatment for AEs as early as within the first month (Figure 3). We found no increase in AEs with longer disease duration. Non-tolerability of the drug seems to become apparent rather early after initiation of treatment.

The rate of 24.3% of patients stopping tofacitinib for adverse events in our study is comparable to the rate of 25% published in the 9.5 year long-term extension study published by Wollenhaupt et al. [52].

### 4.3. Limitations

A significant limitation of this study is that it deals with real-life data. Follow-ups in real-life practice are not as frequent or consistent as in clinical studies. The size of the patient cohort was also limited, and there was a considerable size difference between the sub-cohorts of patients naïve to biologics and patients with prior exposure. However, the design as a retrospective, whole population real-life analysis also constitutes a strength of this study, as its results reflect the variability of patient populations in medical practice more than data on selected patients in controlled clinical trials.

## 5. Conclusions

The efficacy and safety of tofacitinib have been established in clinical trials. This retrospective analysis of real-life data shows that tofacitinib is also effective and safe in a real-life setting. Over 50% of the patient cohort achieved LDA or remission on a dose of tofacitinib 5 mg bid, with a higher rate of patients naïve to biologic agents achieving either LDA or remission. The safety profile of tofacitinib was generally consistent with previous studies. In conclusion, our results support the use of tofacitinib in the treatment of RA to achieve a more successful clinical outcome.

## Figures and Tables

**Figure 1 jcm-08-01548-f001:**
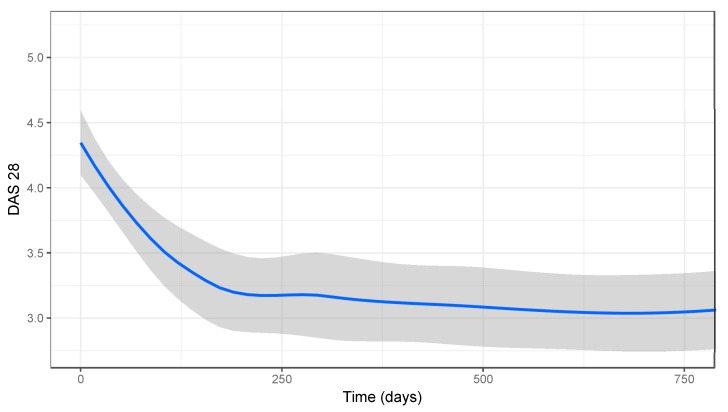
Disease activity: The average disease activity score (DAS28) level is shown for all rheumatoid arthritis (RA) patients treated with tofacitinib with a 95% confidence interval.

**Figure 2 jcm-08-01548-f002:**
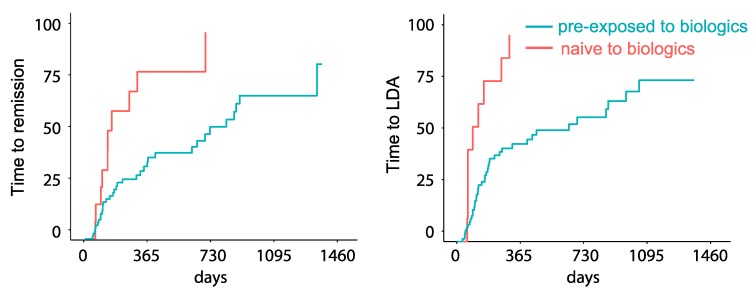
Disease activity: Time to remission (**left panel**) and low disease activity (LDA, **right panel**) is shown for all RA patients treated with tofacitinib. Patients previously exposed to biologic agents are shown in green, and patients naïve to biologics in red.

**Figure 3 jcm-08-01548-f003:**
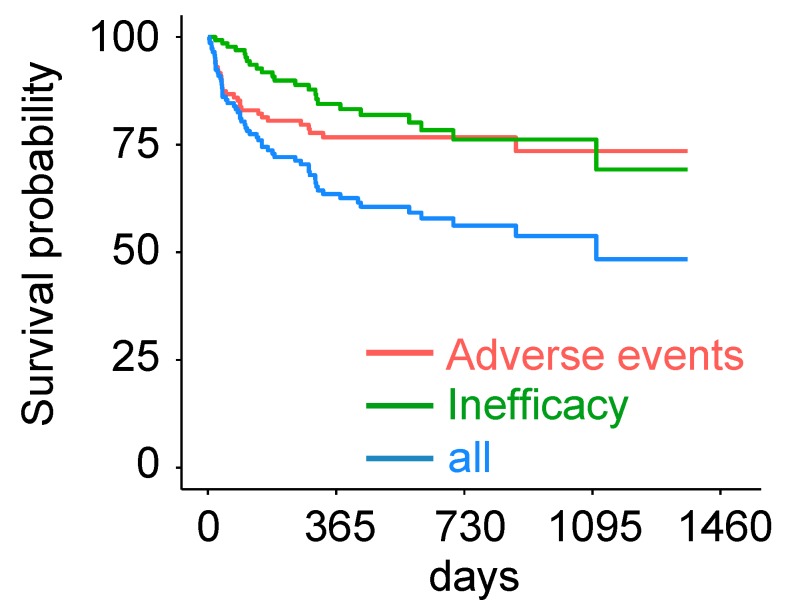
Time to discontinuation of tofacitinib was analyzed for all patients (*n* = 57 out of 144 total patients, blue line). Patients stopping for ineffectiveness (*n* = 22, green line) or adverse events (*n* = 35, red line) are shown separately.

**Figure 4 jcm-08-01548-f004:**
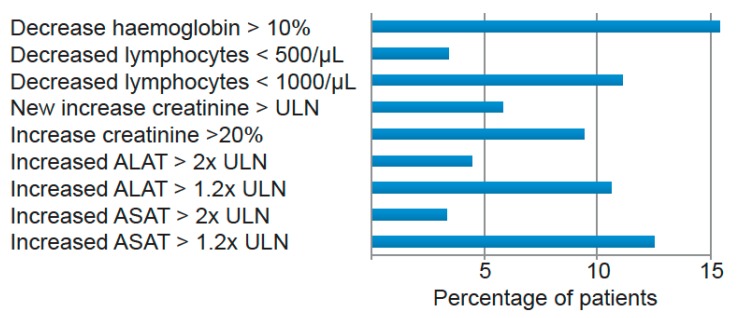
Patients were followed for laboratory changes under treatment with tofacitinib. Data are shown for patients with at least one in- or decrease in one of these parameters during follow-up. Percentages were calculated on patients with available data. ALAT: alanine amino transferase. ASAT: aspartate amino transferase.

**Table 1 jcm-08-01548-t001:** Patient demographics.

	All	Stopped	Remained on	Naïve to	After
	Patients	Tofacitinib		A Biologic Agent	
Number (*n*)	144	57	87	22	122
Gender (%, female)	69.4	64.9	72.4	72.7	68.8
Age at initiation tofacitinib (years, mean)	59.7	59.6	59.8	58.8	59.8
Tofacitinib applied in monotherapy (*n*, %)	65	22	43	14	51
Concomitant medication					
- Methotrexate	36 (25.0)	16 (28.1)	20 (23.0)	5 (22.7)	31 (25.4)
- Sulfasalazine	7 (4.9)	4 (7.0)	3 (3.4)	0 (0)	7 (5.7)
- Leflunomide	25 (17.3)	6 (10.5)	19 (21.8)	3 (13.6)	22 (18.0)
- Hydroxychloroquine	11 (7.6)	5 (8.8)	6 (6.9)	0 (0)	11 (9.0)
- Prednisolone or equivalent	48 (33.3)	16 (28.1)	32 (36.8)	4 (18.2)	44 (36.1)
Disease duration (years, mean)	9.1	9.9	8.7	2.6	10.3
Comorbidities of special interest					
Cardiovascular					
- Coronary heart disease	10	2	8	1	9
- Arterial hypertension	29	13	16	3	26
- Dysipoproteinemia	5	2	3	1	4
- Valvular heart disease	2	1	1	1	1
- Adipositas	12	6	6	0	2
- PAD	3	1	2	0	3
Osteoporosis	39	15	24	4	25
After a biologic agent (%)	84.7	87.2	83.5	0	100
Previous biologic agents (*n*, mean)	2.2	2.3	2.2	0	2.6
Previous csDMARDs (*n*, mean)	1.9	1.9	1.8	1.4	1.9
ACPA pos. (%)	48.6	42.8	52.3	50.0	48.3
Rheumatoid factor pos. (%)	50.0	51.1	48.2	40.9	51.7
Erosive disease (%)	63.3	60.9	66.7	45.5	66.7
DAS28 (mean)	4.4	4.4	4.5	3.7	4.6
ESR (mean)	17.2	18.5	16.6	18.8	16.9
CRP (mean, ULN < 5mg/L)	8.5	8.0	8.8	8.9	8.4

*n*: number. DAS28: 28 joint count disease activity score. DMARDs: disease modifying drugs. ACPA: anti-citrullinated peptide antibody. ESR: erythrocyte sedimentation rate. CRP: C-reactive protein. pos.: positive. ULN: upper limit of normal. PAD: peripheral artery disease.

**Table 2 jcm-08-01548-t002:** Reasons for stopping tofacitinib.

Reason	Number	Time to Stop Tofa
Inefficacy/flare	*n* = 22	median d204, range d21–d1106
Gastrointestinal	*n* = 18	median d28, range d4–d265d
Infection	*n* = 5	median d154, range d85–d877
Myalgia	*n* = 2	range d92–d171
Remission	*n* = 2	range d106–d379
Headache	*n* = 2	d30
Cough	*n* = 1	d22
Blue finger syndrome	*n* = 1	d10
Intolerance	*n* = 1	d42
Heartburn	*n* = 1	d39
Psoriasis	*n* = 1	d287
Increased liver enzymes	*n* = 1	d290

d: day. *n*: number.

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
