# Peer review of "Effectiveness, Tolerability, and Safety of Tofacitinib in Rheumatoid Arthritis: A Retrospective Analysis of Real-World Data from the St. Gallen and Aarau Cohorts"

_jcm, 2019, doi:10.3390/jcm8101548_

Round 1

Reviewer 1 Report

The authors described the Effectiveness, Tolerability, and Safety of Tofacitinib in Rheumatoid Arthritis: Retrospective Analysis of Real-World Data from the St. Gallen and Aarau Cohorts. I think this manuscript is well-written and comprehensive to the readers. The major defect is small sample size real-world study provide nothing new. In the section of 3.4. Laboratory values. Please add full name of the below: aspartate amino transferase/alanine amino transferase : ASAT/ALAT

In the section of 3.3. Discontinuation: None of the demographic parameters was a significant predictor for stopping tofacitinib (data not shown). Please show the data. In the section of 3.2. Disease activity: "Median time to LDA and remission was 319 days and 645 days, respectively (data not shown). " Please show the data. Too many references. Suggest cut to below 30 citations. Figure 3 is confusing, need revision.

Author Response

Respected Reviewer

Thank you for evaluating our manuscript (jcm-271787) entitled: “Effectiveness, tolerability, and safety of tofacitinib in rheumatoid arthritis: A retrospective analysis of real-world data from the St. Gallen and Aarau cohorts” byRuediger B. Mueller, Caroline Hasler, Florian Popp, Frederik Mattow, Mirsada Durmisi, Alexander Souza, Paul Hasler, Andrea Rubbert-Roth, Hendrik Schulze-Koops, and Johannes von Kempis.

we have added a point by point answer to the reviewer’s comments.

Reviewer No. 1:

Reviewer comment: The authors described the Effectiveness, Tolerability, and Safety of Tofacitinib in Rheumatoid Arthritis: Retrospective Analysis of Real-World Data from the St. Gallen and Aarau Cohorts. I think this manuscript is well-written and comprehensive to the readers. The major defect is small sample size real-world study provide nothing new.

Our answer:We agree that sample size is always an issue. The idea was to describe consecutive patients under Tofacitinib. As this was an explorative study there was no power we could calculate in advance. We think that the number analysed is sufficient to draw the conclusions made in the manuscript. The conclusions are clearly supported by the results of the statistical, which demonstrates that the number of cases evaluated was sufficient. Furthermore, the approach we used, i.e. a whole population analysis, precludes biases and renders the results more robust.

Until now mainly clinical trials have been published with tofacitinib. Real life data, however, are important, as the patients in clinical studies were recruited either with a higher disease activity (flair design) or with a better prognosis. Thus, patients who have failed many different bDMARDs are not included in clinical studies.

Naturally, we agree with the reviewer, that a bigger sample size could be desirable, but our analysis covers the largest sample size published with tofacitinib outside of clinical trials and registries. The fact that the statistically significant effectiveness found in our real life dataset is comparable to that in clinical trials is a novel finding, which has not yet been published.

Reviewer comment: In the section of 3.4. Laboratory values. Please add full name of the below: aspartate amino transferase/alanine amino transferase : ASAT/ALAT

Our answer:The spell out for ALAT and ASAT was added in the legend of Figure 4.

Reviewer comment:In the section of 3.3. Discontinuation: None of the demographic parameters was a significant predictor for stopping tofacitinib (data not shown). Please show the data.

Our answer:The wording was misleading. This statement refers to the demographical data at baseline. These data are shown for the whole cohort and the subgroups analysed in table 1. Looking at these data reveals no differences for the subgroups. This was also calculated and no statistical differences were detected. The results of this calculation were not included into the manuscript as the table would not be readable any more. This is the “data not shown”. As this wording misled the respected reviewer, we took the words “data not shown” out of the manuscript in section 3.3.. 

Reviewer comment:In the section of 3.2. Disease activity: "Median time to LDA and remission was 319 days and 645 days, respectively (data not shown). " Please show the data.

Our answer:These data are shown in Figure 2. We think that this figure provides more comprehensive information than a bar graph showing the calculated average times to remission or LDA. Therefore, we would prefer not to show this information in a separate graph and keep the description as it is with naming the average times to LDA and remission. As this wording misled the respected reviewer, we took the words “data not shown” out of the manuscript in section 3.2..

Reviewer comment:Too many references. Suggest cut to below 30 citations.

Our answer:It is a difficult question to decide how many references are enough. Reviewer 1 asks us to reduce the number of references and reviewer 2 asks us to add another three references. We, therefore, kindly ask to leave the number of references as it is.

Reviewer comment:Figure 3 is confusing, need revision.

Our answer:We disagree with the reviewer, that this figure may be confusing. However, the p values referenced in the figure are not anymore included. These p values were still discussed in the figure legend. These sentences were taken out accordingly. Maybe this helps to reduce the confusion about the figure. The details are considered in the text.

Once again, I would like to ask you to reconsider the reviewer’s opinion with regard to our submission and accept this paper for publication in the “Journal of Clinical Medicine” and are looking forward to your answer.

Best,

Ruediger Müller

Reviewer 2 Report

Dear Editor,

I would like to thank you for giving me the opportunity to review the manuscript entitled: Effectiveness, Tolerability, and Safety of Tofacitinib in Rheumatoid Arthritis: a Retrospective Analysis of Real-World Data from the St. Gallen and Aarau Cohorts”.

I have the following points that need to be clarified:

Line 60: the term no-biological DMARDs is not used. I suggest to change it to “conventional synthetic (cs) DMARDs. Line 61: The authors are stating that the effectiveness of bDMARDs is limited. 50% of RA patients treated with bDMARDs meet the criteria of remission or LDA and that the bDMARDs may present adverse events (line 63-67). Therefore, there are unmet needs in RA management, thus the use of tofacitinib is an alternative treatment (line 68-69). Lines 79-82: On the other hand, they are stating that studies comparing tofacitinib to other bDMARDs have similar results regarding effectiveness, safety and tolerability.

The sentences in points 2 and 3 sound somewhat controversial. If the results for tofacitinib vs. the other bDMARDs in the RCT are similar regarding safety and efficacy, then why the authors suggest to use tofacitinib in open label fashion and in a retrospective analysis?

I think that the results of the present retrospective study are quite similar to the RCT. In addition, a significant number of patients (22,2%) discontinued the study due to adverse events, while 14.6% stopped receiving tofacitinib due to inefficacy.

The disease duration, as it is depicted in Table 1, is 9.1 years, while the disease was erosive in 63.3% of the patients. In addition, the presence of RF and ACPA were approximately 50%. How do the authors explain all the above? In the same table no other drugs in addition to tofacitinib are shown. I suppose that the patients received tofacitinib as monotherapy, without any csDMARDs or small doses of steroids. Is this correct? The authors claim that this is a real-word data study. If so, in table 1 there are many important missing parameters. For example: BMI, smoking status, other comorbidities (diabetes mellitus, hypertension etc) or any other concomitant drugs used. The DAS 28 on the initiation of tofacitinib was 4,4 and decreased to 3.13 (figure 1). This means a reduction of 1.27 which means a moderate disease activity according to the EULAR criteria. In line 230 the authors are stating that 32 patients (22.2%) stopped receiving tofacitinib due to adverse events, while in the discussion line 517, they are stating that the rate of adverse events was 24.4%. Which is true? The bibliography reported in lines 518-519 is not listed in the reference list. There are several recent important papers regarding RA treatment.

I suggest to add and discuss appropriately the following:

1 J Clin Med 2019;8:E938, doi 103390/jcm 8070938

J Clin Med 2019;8:E1237 doi: 10.3390/jcm8081237 Sem Arthritis Rheun 2019;48:597-602

Author Response

Respected reviewer

Thank you for evaluating our manuscript (jcm-271787) entitled: “Effectiveness, tolerability, and safety of tofacitinib in rheumatoid arthritis: A retrospective analysis of real-world data from the St. Gallen and Aarau cohorts” byRuediger B. Mueller, Caroline Hasler, Florian Popp, Frederik Mattow, Mirsada Durmisi, Alexander Souza, Paul Hasler, Andrea Rubbert-Roth, Hendrik Schulze-Koops, and Johannes von Kempis.

We have added a point by point answer to the reviewer’s comments.

Reviewer No. 2:

Reviewer comment: Line 60: the term no-biological DMARDs is not used. I suggest to change it to “conventional synthetic (cs) DMARDs.

Our answer:The term Non-biological was exchanged forconventional synthetic.

Reviewer comment: Line 61: The authors are stating that the effectiveness of bDMARDs is limited. 50% of RA patients treated with bDMARDs meet the criteria of remission or LDA and that the bDMARDs may present adverse events (line 63-67). Therefore, there are unmet needs in RA management, thus the use of tofacitinib is an alternative treatment (line 68-69). Lines 79-82: On the other hand, they are stating that studies comparing tofacitinib to other bDMARDs have similar results regarding effectiveness, safety and tolerability. The sentences in points 2 and 3 sound somewhat controversial. If the results for tofacitinib vs. the other bDMARDs in the RCT are similar regarding safety and efficacy, then why the authors suggest to use tofacitinib in open label fashion and in a retrospective analysis?

Our answer:On one hand there is data from clinical trials on a selected population of patients. This selected population of patients does not necessarily reflect the situation in real life. Therefore, as stated in the last paragraph of the introduction “there is a demand to use the experience gained through using tofacitinib in a real-life, clinical setting, to further evaluate its safety and utility. In this study, we aimed to analyze real-life data from routine clinical practice to compare our experience with the results of controlled studies“. The reason to introduce new thearpies is the failure of existing therapeutics to improve disease activity and to provide alternatives when they fail or are not tolerated.

Reviewer comment: I think that the results of the present retrospective study are quite similar to the RCT. In addition, a significant number of patients (22,2%) discontinued the study due to adverse events, while 14.6% stopped receiving tofacitinib due to inefficacy.

The disease duration, as it is depicted in Table 1, is 9.1 years, while the disease was erosive in 63.3% of the patients. In addition, the presence of RF and ACPA were approximately 50%. How do the authors explain all the above?

Our answer:There is no other explanation than “this is the data of the patients who were identified for this whole population analysis”. As mentioned for the last question, patients included in a clinical study have always a highly selected demographical pattern, perhaps optimised for a potential response to a new drug. Patients being initiated on a new drug, e.g. tofacitinib, are not necessarily patients the pharmaceutical companies would select for a clinical study. Therefore, these data differ from the data seen in clinical studies. Despite the high percentage of erosive disease, the data were in line with previous registration trials.

Reviewer comment: In the same table no other drugs in addition to tofacitinib are shown. I suppose that the patients received tofacitinib as monotherapy, without any csDMARDs or small doses of steroids. Is this correct?

Our answer:The number of patients on tofacitinib monotherapy without concomitant csDMARDs was added in table 1 per group.

Reviewer comment: The authors claim that this is a real-word data study. If so, in table 1 there are many important missing parameters. For example: BMI, smoking status, other comorbidities (diabetes mellitus, hypertension etc) or any other concomitant drugs used.

Our answer:The reviewer is right: there are missing data on BMI, smoking status, and comorbidities or other concomitant drugs. For some, we had incomplete data (BMI, smoking status), others(diabetes mellitus, hypertension) were left out to focus the paper on efficacy and reasons to stop tofacitinib.

Reviewer comment: The DAS 28 on the initiation of tofacitinib was 4,4 and decreased to 3.13 (figure 1). This means a reduction of 1.27 which means a moderate disease activity according to the EULAR criteria.

Our answer: According to J. Fransen (Clin Exp Rheumatol 2005; 23 (Suppl. 39):S93-S99) a good EULAR response is achieving a DAS 28 ≤ 3.2 and a change of DAS of > 1.2. Thus, this good EULAR response was achieved in our patient group, on average.

Reviewer comment: In line 230 the authors are stating that 32 patients (22.2%) stopped receiving tofacitinib due to adverse events, while in the discussion line 517, they are stating that the rate of adverse events was 24.4%. Which is true? The bibliography reported in lines 518-519 is not listed in the reference list.

Our answer:We thank the reviewer for detecting this mishap. In total 35 patients stopped for AE, respectively 24.3%. All numbers were revised.

Reviewer comment: There are several recent important papers regarding RA treatment. I suggest to add and discuss appropriately the following:

J Clin Med 2019;8:E938, doi 103390/jcm 8070938

J Clin Med 2019;8:E1237 doi: 10.3390/jcm8081237

Sem Arthritis Rheun 2019;48:597-602

Please not that I still have a free waver for this manuscript.

Our answer:The three papers are now referenced in the 3rd, 1st, and 2ndparagraph of the introduction, respectively.

Once again, I would like to ask you to reconsider the reviewer’s opinion with regard to our submission and accept this paper for publication in the “Journal of Clinical Medicine” and are looking forward to your answer.

Best,

Ruediger Müller

Round 2

Reviewer 1 Report

 No further comment 

Author Response

Thank you

Reviewer 2 Report

Dear Editor

Thank you very much for sending us the manuscript entitled “EFFECTIVENESS, TOLERABILITY, AND SAFETY OF TOFACITINIB IN RHEUMATOID ARTHRITIS: A RETROSPECTIVE ANALYSIS OF REAL-WORLD DATA FROM THE ST. GALLEN AND AARAU COHORTS”  for revision.

The authors have responded to some of my queries. However, despite that they have added the 3 suggested references, the authors neither discussed nor appropriately commented any of them on their manuscript. In addition, Table 1 is still incomplete. Demographic data such us BMI, smoking status, comorbidities and other concomitant drugs are missing.

Author Response

Thank you for evaluating our manuscript (jcm-271787) entitled: “Effectiveness, tolerability, and safety of tofacitinib in rheumatoid arthritis: A retrospective analysis of real-world data from the St. Gallen and Aarau cohorts” byRuediger B. Mueller, Caroline Hasler, Florian Popp, Frederik Mattow, Mirsada Durmisi, Alexander Souza, Paul Hasler, Andrea Rubbert-Roth, Hendrik Schulze-Koops, and Johannes von Kempis.

We have added a point by point answer to the 2nd reviewer’s comments, as reviewer No. 1 had no further comments.

Reviewer No. 2:

Reviewer comment: The authors have responded to some of my queries. However, despite that they have added the 3 suggested references, the authors neither discussed nor appropriately commented any of them on their manuscript.

Our answer: The requested references were included in the introduction:

J Clin Med 2019;8:E938, doi 103390/jcm 8070938is an in-depth analysis of real-world patient data that addresses therapeutic strategy in RA with the aim of quantitating the proportion of patients with insufficient responses to csDMARDs and bDMARDs. The concluding statements “intervention must be: early and correct diagnosis, T2T approach following the ACR/EULAR recommendations and strategies, as well as, close follow-up and monitoring” or the arising matters “Is there indeed such gap and huge unmet needs in RA management? Are the data from registries and expert opinions strong enough to suggest such high unmet needs for RA management? Are we still in need of more long-term and better structured observational studies to clarify the subject? On the other hand, do all physicians follow the ACR/EULAR recommendations for RA management and the treat-to-target (T2T) approach? Does training and education in rheumatology differ among countries?” are not directly addressed by this paper. The only question “Are the data from registries and expert opinions strong enough to suggest such high unmet needs for RA management?” which could be addressed in this paper is a rather a philosophical question. We have discussed the limitations and importance of real life data in paragraph 4.3..

The study by Drosos et al. was referenced in the introduction, where the therapeutic goals and strategies are discussed, as we consider this paper to be important in the field.

J Clin Med 2019;8:E1237 doi: 10.3390/jcm8081237is a general review paper on the therapeutic strategy in RA with no direct relation to the real life data. Therefore it was referenced in the introduction, where the therapeutic goals and strategies are discussed.

Sem Arthritis Rheum 2019;48:597-602describes a treat to target approach using csDMARD in an observational cohort of early RA patients. We think it is an important paper. The aim of our study was to analyse tofacitinib treatment in an observational cohort. None of the patients was treated early or to target (T2T) for this analysis. Thus, the analysis presented is, apart from the fact that both analyses are observational cohorts in RA, completely different. As a consequence, we have not discussed these data, but only referenced them in the introduction.

Reviewer comment: In addition, Table 1 is still incomplete. Demographic data such us BMI, smoking status, comorbidities and other concomitant drugs are missing.

Our answer:For the missing data in table 1 we would like to give the same answer again as the last time: The reviewer is right: there are missing data on BMI, smoking status, and comorbidities or other concomitant drugs. For some, we had incomplete data (BMI, smoking status), others (diabetes mellitus, hypertension) were left out to focus the paper on efficacy and reasons to stop tofacitinib.

Once again, I would like to ask you to reconsider the reviewer’s opinion with regard to our submission and accept this paper for publication in the “Journal of Clinical Medicine” and are looking forward to your answer.

With my best regards,

Ruediger Müller

This manuscript is a resubmission of an earlier submission. The following is a list of the peer review reports and author responses from that submission.

Round 1

Reviewer 1 Report

This retrospective study can be improved by active comparators, eg other biologics, and should be followed for longer duration and larger sample size.

Author Response

We agree with the reviewer that a correlation with active comparators, a longer duration, and a bigger sample size would be interesting. The number of patients in this manuscript, however, is the largest number of patients treated with tofacitinib in a real world cohort published outside of RCT and registries so far. Looking at our data, we doubt that a longer follow up or an increased sample size may add important information or increase the clinical significance of these data.

Reviewer 2 Report

The paper reports data from real-life analysis of RA patients under tofacitinib, and this is its strength (as well as its weakness compared to RCTs). This reviewer strongly believes that data stemming from medical practice, especially those investigating the safety profile of biologics used for the treatment of an autoimmune disease, such as RA, are very important.

Comments

The authors must clarify whether percentages of RF and anti-CCP are those reported in the records of patients or those prior to tofacitinib treatment. How many of the patients enrolled were diagnosed as 'seronegative RA' and how many as seropositive? How many had RF/and or anti-CCP?

All values which are given as median must include ranges (and these must include also the median of adverse reactions). 

Lines 231. authors report Hb decrease<10%. they must clarify the respective % of Hb on tafacitinib versus non-tofacitinib.

Figure 4 does not project well.

The paper's text includes several changes indicated on the right of the manuscript but this Reviewer is unable to trace whether such changes are made by authors as a response to a Reviewer or any other reason during the preparation of the manuscript. 

if values are indicated (for example CRP), normal values/cut-offs must be given

Author Response

Reviewer’s comment:The authors must clarify whether percentages of RF and anti-CCP are those reported in the records of patients or those prior to tofacitinib treatment. How many of the patients enrolled were diagnosed as 'seronegative RA' and how many as seropositive? How many had RF/and or anti-CCP?

Our answer:In section 3.1. an explanatory comment that RF and ACPA were taken from the files and no additional testing for RF and/or ACPA were conducted before or under treatment with tofacitinib. 56% of the patients were either RF and/or ACPA pos. This information was added to the manuscript in section 3.1.

Reviewer’s comment:All values which are given as median must include ranges (and these must include also the median of adverse reactions). 

Our answer:Ranges were included into the table

Reviewer’s comment:Lines 231. authors report Hb decrease<10%. they must clarify the respective % of Hb on tafacitinib versus non-tofacitinib.

Our answer:There were no patients without tofacitinib in this analysis.

Reviewer’s comment:Figure 4 does not project well.

Our answer:A new version of figure 4 was integrated into the manuscript.

Reviewer’s comment:The paper's text includes several changes indicated on the right of the manuscript but this Reviewer is unable to trace whether such changes are made by authors as a response to a Reviewer or any other reason during the preparation of the manuscript. 

Our answer:We agree that a track changes version of a reviewed article is important for the reviewer to see, the changes made under reviewing. The manuscript was reviewed intensively for the English language with native speakers and, secondly, we tried to improve the readability of the manuscript. Therefore many changes were included, which did not directly correlate with the reviewers comments. This general revision was a request asked by the reviewers of the first revision.

Reviewer’s comment:if values are indicated (for example CRP), normal values/cut-offs must be given

Our answer:The upper limit of normal for CRP was included into the manuscript.

Round 2

Reviewer 1 Report

Open label case series are of less informative on evidence.